# Ultrasound Axicon: Systematic Approach to Optimize Focusing Resolution through Human Skull Bone

**DOI:** 10.3390/ma12203433

**Published:** 2019-10-20

**Authors:** Fabián Acquaticci, Sergio E. Lew, Sergio N. Gwirc

**Affiliations:** 1Instituto de Ingeniería Biomédica, Universidad de Buenos Aires, Buenos Aires C1063ACV, Argentina; slew@fi.uba.ar; 2Instituto Nacional de Tecnología Industrial, Ministerio de Producción y Trabajo, San Martín, Buenos Aires B1650WAB, Argentina; 3Departamento de Investigaciones Tecnológicas, Universidad Nacional de La Matanza, San Justo, Buenos Aires B1754JEC, Argentina

**Keywords:** ultrasonic lens, axicon lens, focused ultrasound, transcranial ultrasound

## Abstract

The use of axicon lenses is useful in many high-resolution-focused ultrasound applications, such as mapping, detection, and have recently been extended to ultrasonic brain therapies. However, in order to achieve high spatial resolution with an axicon lens, it is necessary to adjust the separation, called stand-off (δ), between a conventional transducer and the lens attached to it. Comprehensive ultrasound simulations, using the open-source k-Wave toolbox, were performed for an axicon lens attached to a piezo-disc type transducer with a radius of 14 mm, and a frequency of about 0.5 MHz, that is within the range of optimal frequencies for transcranial transmission. The materials properties were measured, and the lens geometry was modelled. Hydrophone measurements were performed through a human skull phantom. We obtained an initial easygoing design model for the lens angle and optimal stand-off using relatively simple formulas. The skull is not an obstacle for focusing of ultrasound with optimized axicon lenses that achieve an identical resolution to spherical transducers, but with the advantage that the focusing distance is shortened. An adequate stand-off improves the lateral resolution of the acoustic beam by approximately 50%. The approach proposed provides an effective way of designing polydimethylsiloxane (PDMS)-based axicon lenses equipped transducers.

## 1. Introduction

Focused ultrasound (FUS) in pulsed mode is unique among transcranial brain stimulation methods in combining exceptional spatial resolution (on the millimeter scale) [1,2,3] with the potential to target subcortical structures (deeper than 10 cm) [4] through the intact skull. It also has potential for inducing neuronal excitation or suppression without evidence of tissue damage [5].

Recently, we demonstrated the advantages of focusing ultrasound (US) through polydimethylsiloxane (PDMS)-based axicon lenses to selectively drive brain activity [6]. The ultrasound axicon is shown in Figure 1. It has the shape of a cone. As the cone angle (*φ*) decreases, the focus moves closer to the lens. Developing low-intensity applications include the opening of the blood–brain barrier and ultrasonic neuromodulation. Both techniques have recently been extended to human subjects and are under active research. Spherically FUS transducer has been, until now, the most commonly used for transcranial focusing ultrasound, but it has a large focal length (several centimeters) that may hinder its coupling with the head, for example, when the focal plane is too close to skull inner ivory layer. With spherical segment ultrasound transducers, FUS is commonly delivered through a big plastic bag containing degassed water placed over the scalp. This is due to the fact that, for cortical stimulation, the acoustic beam should be focused a few millimeters deep from the skull surface [3]. In this sense, the great advantage of the axicon lens is its ability to suppress the near-field and maintain a very near focus from the lens face outward. PDMS-based axicon lens affixed on the face of conventional transducers makes it possible to build more compact devices without liquids that can regasify or leak, with a greater spatial resolution.

However, for an axicon lens to offer high spatial resolution and depth, control is necessary to adjust the separation between the transducer and axicon lens, called stand-off (*δ*), for each particular case [7]. Given the geometry of the lens, if *δ* is not properly adjusted, internal reflections can occur, making the configuration useless as a focused transducer. The lens reduces the focal length (F) from the near-field distance (N) of the attached US transducer. The relation between axicon lens angle and the value of F/N produced was described in [8] for acrylic plastic/oil combination, but the effect of the stand-off was not modeled and the adequate value of *δ* was not described elsewhere. In [7], the stand-off of the different settings was adjusted experimentally to obtain the best signal-to-noise ratio (SNR) for ultrasonic contact inspection. In our approach, time domain ultrasound simulations, based on the k-space pseudo-spectral methods, play a key role in enabling the modeling and systematic design of ultrasonic axicon lenses with optimum stand-off for high-resolution ultrasonic applications, as transcranial stimulation, in high-resolution mapping, and in the evaluation of a wide variety of defects. Finally, the effects of human skull on axicon fields were tested.

Skull is generally constituted of three relatively homogeneous layers: The outer and inner tables of solid ivory bone, and the central layer of diploe of cancellous bone, with a blood- and fat-filled porous structure. The dimensions of the blood- and fat-filled inclusions are random, and an average thickness in the direction normal to the surface is about 0.6 mm. [5], so the fundamental frequency used was chosen to help alleviate the concerns for acoustic energy absorption or refraction by the skull. For frequencies lower than about 0.5 MHz, reflection of sound is the principal cause of insertion loss. At frequencies between about 0.6 and 0.9 MHz, the absorption loss in the diploe layer begins to limit sound transmission, so the oscillations are damped out and the insertion loss increases linearly with frequency. At about 0.9 MHz, the scattering loss in the diploe layer begins to limit sound transmission, so the insertion loss begins to increase as the fourth power of frequency [5].

A full-wave nonlinear ultrasound model based on the k-space pseudo-spectral method was developed and released as part of the open-source k-Wave Acoustics Toolbox [9,10]. This model can account for the propagation of nonlinear ultrasound waves in homogeneous or heterogeneous media with power acoustic absorption and without restrictions on the directionality of the waves. The accuracy of the implementation of nonlinear ultrasound model in the k-Wave Toolbox was validated using experimental measurements of the ultrasound made with a linear diagnostic ultrasound probe and a membrane hydrophone [11].

A computational model for elastic wave propagation in heterogeneous media can be constructed based on the solution of coupled first-order acoustic equations given in Equations (1)–(3) using the Fourier pseudo-spectral method. This uses the Fourier collocation spectral method to compute spatial derivatives, and a leapfrog finite-difference scheme to integrate forwards in time. Using a temporally and spatially staggered grid, the field variables are updated in a time stepping [12,13]. To simulate free-field conditions, a perfectly matched layer (PML) is also applied to absorb the waves at the edge of the computational domain [14]. Without this boundary layer, the computation of the spatial derivate via the FFT causes waves leaving one side of the domain to reappear at the opposite side. The use of the PML thus facilitates infinite domain simulations without the need to increase the size of the computational grid.
(1)$$\frac{\partial u}{\partial t}=-\frac{1}{\rho_{0}}\nabla p-\alpha \cdot u$$
(2)$$\frac{\partial \rho_{x}}{\partial t}=-\rho_{0}\frac{\partial u_{x}}{\partial x}-\alpha_{x}\rho_{x}$$
(3)$$p=c_{0}^{2}\sum^{\;}\rho_{x,y,z}$$
where *u* is the acoustic particle velocity, *ρ_0_* is the medium density, *ρ* is the acoustic density, *c* is the thermodynamic sound speed, p is the acoustic pressure, and p_0_ = p(*t* = 0) is the initial pressure distribution.

There are two main stages in this work: The definition of general design equations for PDMS-based axicon lenses with optimal stand-off, through comprehensive ultrasound simulations, for an optimal transcranial focusing; and the measurements and simulations performed to determine focusing performance of the proposed lenses, through a human skull phantom.

## 2. Materials and Methods

In order to develop a design model, axicon lenses with *φ* angles between 80° and 170° in steps of 5° were simulated with increments of *δ* in steps of *λ*/4 with 8 grid points per wavelength (λ) in the stand-off medium. The final pressure field along with the RMS beam pattern was calculated. The normalized cross-section profile area at the focus F was determined, for each δ. The minimum area represents the greatest improvement with the optimum stand-off, which minimizes transmitted energy outside of the main beam and improves lateral spatial resolution with lower possible sidelobes.

The domain was discretized using a grid point spacing of 250 μm (giving a maximum supported frequency of 2.06 MHz), and a grid size of 512 × 512 grid points (corresponding to a domain size of 128 × 128 mm). Simulations were run on an NVIDIA^®^ GTX 950 graphics processing unit (Santa Clara, CA, United States) using the MATLAB^®^ Parallel Computing Toolbox (Natick, MA, United States). The simulation of all angles/stand-off combinations can be completed in approximately 60 h. By default, numbers in MATLAB^®^ are stored in double precision. However, in almost all cases, k-Wave does not require this level of precision. In particular, the performance of the PML generally limits the accuracy to around 4 or 5 decimal places. We use a PML thickness of 20 grid points that gives a transmission coefficient of −100 dB. This corresponds to a reduction in the signal level of 1 × 10^−5^, which is significantly less than double precision. Further, there will also be uncertainties in the definition of the materials properties. A list of the main simulation inputs is given in Table A1. In this work, a heterogeneous medium was defined as a layered interface on both sides of the conical cavity of the axicon lens as shown in Figure A1. The convective nonlinear effects from the convection of mass was considered. However, at low frequencies and amplitudes, nonlinearity will only have a small effect on the wave field. At higher frequencies and amplitudes, this effect become more important.

The accuracy of the implementation of ultrasound model with the k-Wave Toolbox was validated in our previous work [6] using experimental measurements of FUS made with the same axicon lens attached transducer and a needle hydrophone (Force Technology MH28) within a 6-L anechoic test tank. Our previous study has already shown that there is a good agreement between the simulated and experimental beam patterns. In this work, we also characterized the acoustic pressure amplitude of the beam pattern of the axicon lens when FUS was transmitted through a human skull phantom for experimental validation of simulated transcranial ultrasound propagation. There is negligible conversion to shear waves in the layers of skull when the incidence angle is within about 20° of normal [5]. The ability of bone to support shear waves can affect transcranial transmission, although the changes to the intracranial field are typically negligible for ultrasound applied at normal or near-normal incidence. Therefore, we will model only longitudinal waves. The phantom was created from the parietal portion of a mesh segmented from MRI head image data. Clear Med610 3D printing resin was used to create the skull bone phantom. The acoustic properties of Stratasys™ materials were recently reported in [15]; thus, these measurements were not repeated as part of the current work. The reported and measured property values, and estimated uncertainty in those measurements, are shown in Table 1.

To test the effects of a human skull on FUS fields, we inserted a 5 mm thick fragment of parietal bone phantom between the transducer and the hydrophone, as shown in Figure 2. The transducer described in our previous work [6] has an ultrasonic piezo-disc-type element of 28 mm diameter (SMD28T21F1000R, Steminc Steiner & Martins, Inc., Davenport, FL, USA) of PZT-4 mounted on stainless-steel housing operating in thickness mode vibration at 445 kHz. Epoxy (Resoltech 1040, Resoltech, Rousset, France) resin was used in order to build the conical cavity of the lens with an angle *φ* of 144°. Degassed PDMS (Sylgard 184, Dow Corning, Midland, MI, USA) was used to fill the conical cavity of the lens and for the lens-transducer interface with a stand-off *δ* of 30 mm. The ultrasonic lens is formed by the epoxy/PDMS interface. The specifications are summarized in Table 2.

## 3. Results

### 3.1. Estimation of the Axicon Lens Angle

The relation between axicon lens angle *φ* and the value of F/N produced (see Appendix A) is illustrated graphically in Figure 3. The transducer near field length N is given by *N* = *D*^2^*f*/4*c*. The coefficient of determination R^2^ is 0.97 with *p*-value significance level of < 0.00001. This relation is for *δ* = *δ*_Optimum_:
(4)$$\phi =\frac{9.9708+\ln\left(\frac{F}{N}\right)}{5.52\cdot 10^{-2}}\;\left[degrees\right].$$

The following relation, based on our study summarized in Table A2 (see Appendix A), was found experimentally valid for the lens described. The optimum ratio of F/N appears to be between 0.1 and 0.3. This produces focal beam diameters (d_F_ with the axicon lens equipped transducers and d_N_ of a conventional transducer), and depths of focus (DOF) of similar ratio according to:(5)$$\frac{F}{N}=\frac{d_{F}}{d_{N}}=\frac{{\mathrm{DOF}}_{F}}{{\mathrm{DOF}}_{N}}.$$

For values of F/N > 0.4, some evidence of the original near field still remains. As the value F/N decreases below 0.4, all evidence of the original near field is rapidly suppressed in the lens system. This contrasts with the behavior of spherical lenses where some original near field is always present.

We observe that there is an inversely proportional relationship between F/N and *κδ* values, as shown in Figure 4 for different values of angle *φ*. The relative percentage loss of lateral resolution (LLR) as a function of *κδ* is shown in the same figure, where *κ* is the wave number. LLR is relative to the best lateral resolution that can be achieved with *κδ* value that minimizes energy transmitted outside of the narrowest possible main beam. Since the speed of sound in PDMS is lower than in water, as the value of *δ* increases, the wavelengths-weighted average sound speed through the heterogeneous medium with step discontinuity of velocity decreases, which effectively reduces the ratio of F/N.

### 3.2. Reflectivity Effect on the Internal Walls of the Lens Housing

To illustrate the effect of outer case and inner isolation (Figure 5) on the lateral resolution of the lens, the relative percentage LLR as a function of *κ**δ,* and the normalized lateral beam profile are shown in Figure 6 for two different housing. With a reflecting housing, without inner sleeve, there will be more internal reflections in the lens system, since there are abrupt transitions of acoustic impedance with the inner walls. As a result, the lateral resolution of the lens decreases, compared to non-reflective housing with inner isolation. For example, with a PDMS-based lens at 445 kHz and a cone angle of 130°, assembled with the optimum stand-off inside a housing of Inconel-625 with internal reflectivity of 0.8 dB down, the relative LLR value is reduced by half and energy transmitted outside the main beam is 10% higher, compared to inner sleeve reflectivity of 40 dB down.

### 3.3. Estimation of the Optimum Stand-Off

We found a numerical relationship for *δ* based on the simulation results of 2420 angle/stand-off combinations for axicon lenses (see Appendix A). The relation between the optimum value of *δ* (which improves spatial resolution) and the value of F/N is shown in Figure 7. The linear regression equation is given by:(6)$$\delta_{Optimum}=\frac{10.921+\ln\left(\frac{F}{N}\right)}{1.96\cdot 10^{2}}\;\left[meters\right].$$

The coefficient of determination R^2^ is 0.96 with a *p*-value significance level of < 0.00001.

As an example, Figure 8 shows the focusing behavior of PDMS-based 144° axicon lens with four different values of *δ* (a: 0 mm, b: 20.5 mm, c: 34 mm, and d: 45.25 mm). These different settings are indicated in Figure 9 showing the relative LLR as a function of *κ**δ.* With a stand-off of 34 mm, the lateral spatial resolution improves by up to 40%, compared to the same lens without stand-off.

The optimum stand-off predicted by Equation (6) was checked for different frequencies. Figure 10 compares the experimental loss of lateral resolution (LLR) for transducers with acoustic frequencies (f) of (a) 0.2225 MHz, (b) 0.445 MHz, (c) 0.890 MHz, and (d) 4.45 MHz.

### 3.4. Acoustic-Field Experimental Scan

The hydrophone scans were performed both without and through the phantom. For the scan with the skull, the starting distance to the transducer was increased to 10 mm to avoid collision between the skull and hydrophone. Experimental beam patterns produced are shown in Figure 11. The lateral dimension of FUS beam cross-profiles measured at −6 dB drop of the pressure at the focus was 3.5 mm in the free space condition and 5 mm after transcranial transmission. We also characterized the acoustic field in the axial direction, perpendicular to the lens face and skull. The FUS pressure half width of the half maximum was 22 mm in the free space condition and 18 mm after transcranial transmission. Under these conditions, transmission of 445 kHz FUS-axicon lens through the skull led to an approximately 40% loss in lateral resolution of the acoustic beam, and on the other hand, an approximately 18% increase in the axial resolution. When FUS was transmitted through the skull, the acoustic pressure dropped by half. The insertion loss of our skull phantom was approximately −6 dB.

Intracranial focal characteristics obtained by simulation using the same configuration of Table 2 for different thicknesses of the skull are indicated in Table 3. There is a good coincidence between the simulation and the values obtained experimentally in water for the skull thickness of 5 mm, separated 2 mm from the lens, as shown in Table 4.

## 4. Discussion

Although the analysis was carried out with a single element transducer with no aberration correction, and a specific skull geometry designed to approximate the varied shaped of the skull, it is expected that the relative influence of different medium properties and aspects of medium geometry will be maintained. Phantom geometry is the major material influence on the intracranial field and sound speed is shown to be the most influential acoustic property in focus pressure, position, and volume. From the experimental beam patterns shown in Figure 11, the unexpected focusing property of the skull in axial axis may be described as a nonlinear effect that causes the beam to rotate back toward the skull insertion point, creating a more compact pressure cigar-shaped acoustic field. This was also observed using segmented-sphere transducers [16,17]. Thus, the skull is not an obstacle for transcranial focusing of US and may exert an additional acoustic lensing effect to enhance spatial resolution under certain conditions. Out of focus, the sound pressure decreases with a very steep slope. From the comparison in Table 4, there is a good coincidence between the simulation and the values obtained experimentally in water for skull thickness of 5 mm, separated 2 mm from the lens. Regarding the influence of the thickness skull in the focus of axicon lenses, since on both sides of the skull there is a discontinuity in the acoustic impedance, with different velocities of sound propagation, and the average sound speed is the acoustic property that most influences the focal distance, we observe as expected, small variations in the position, diameter, and depth of focus, for different thicknesses of the cranial bone, resulting in the average deviations of the focus less than 1 mm.

On the other hand, transmitting FUS through human cranial bone caused an approximately 40% loss in lateral resolution of the acoustic beam, estimated by the intensity full width at half maximum. However, this loss of resolution is compensated by adequate stand-off of axicon lens. In addition, we find that the PDMS provides a smaller focal zone, which is desirable for neurostimulation [6]. All this allows a higher resolution, comparable to spherical transducers (the most commonly used for ultrasonic brain therapy), but with the advantage that the near field is eliminated, and the focus distance is shortened. For other applications, a wider focal zone and a line focus such as that obtained with glycerin or ethylene glycol may be desirable [7]. From Equation (5), the elimination of the near field, by means of an appropriate axicon lens, enables transducers featuring the same wavelength/diameter ratio to produce the same focal spot size. Thus, large diameter, low-frequency transducers may be used. This is useful for brain stimulation where low frequencies are required for penetration of the skull.

One problem of devices with axicon lenses are the relatively high sidelobes [18]. How much this will affect will depend on the proposed applications. With the calculation of the optimum value of *δ*, by Equation (6), a better lateral resolution is achieved. This relation between axicon lens stand-off and the value of F/N is applicable to high-resolution epoxy resin/PDMS lenses or other similar combination.

## 5. Conclusions

The numerical approach proposed in this paper provides a complete and effective way of designing axicon lenses for many high-resolution applications, such as mapping or detection. This is also suitable for focused ultrasound through human skull bone. In view of providing better lateral resolution with lower sidelobes, the use of design programs for this task is not as straightforward. The choice of a good starting point is an important factor for successful optimization. It is easier to obtain a starting design using relatively simple formulas and then use it in a lens design program for future analysis and optimization. We believe that this will be an effective way of designing axicon lenses, for example, to build focused windows to the brain for clinically viable transparent cranial implant for chronic ultrasonic therapy and stimulation of the brain.

## Figures and Tables

**Figure 1 materials-12-03433-f001:**
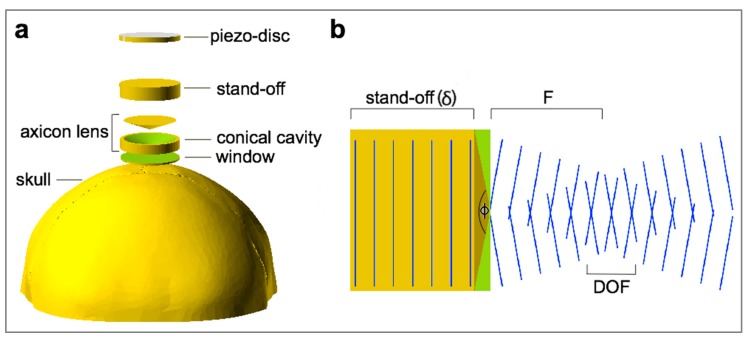
(**a**) Exploded model of the axicon lens. The polydimethylsiloxane (PDMS) fills the conical cavity of the lens and the stand-off. An ultrasonic lens is formed by the PDMS/plastic interfaces. The window is a thin material layer between the conical cavity and the face of the lens; (**b**) schematic ultrasonic beam pattern for transducers with axicon lenses. These are described by its total included cone angle (*φ*). Depth of focus (DOF) is the focal region and F is the desired focal length. Both DOF and F depend on the sound velocity into the material used for the interface.

**Figure 2 materials-12-03433-f002:**
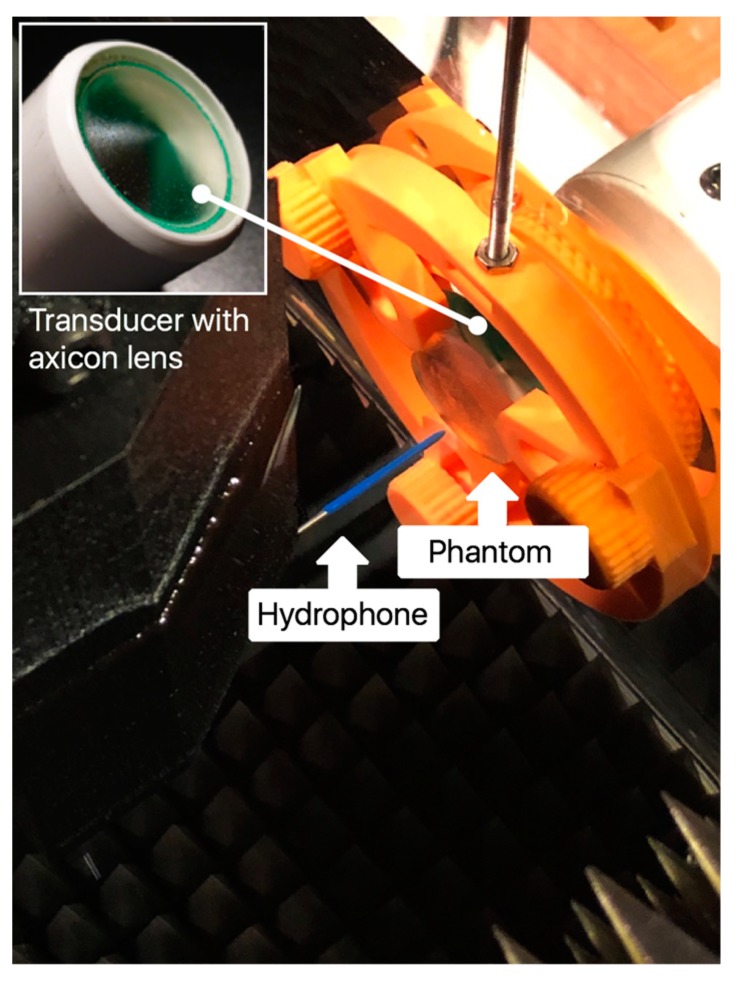
Photograph of the ultrasound test tank showing the axicon lens equipped transducer detail, parietal bone phantom, and hydrophone.

**Figure 3 materials-12-03433-f003:**
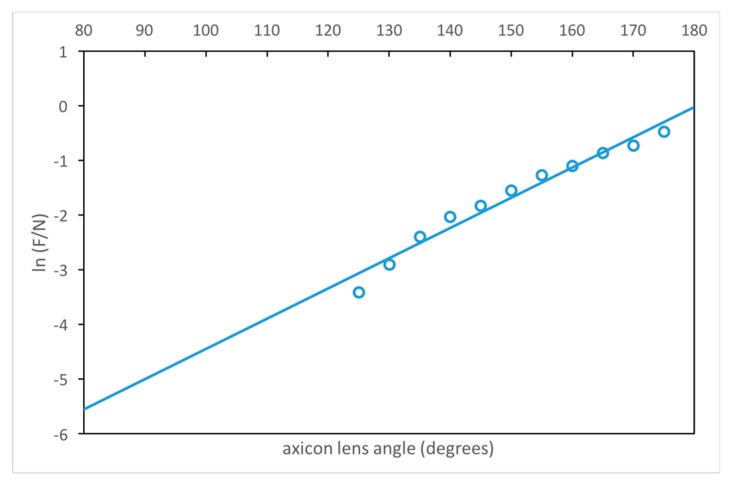
Illustration of the relation between the axicon lens angle *φ* and the ratio of *F/N*.

**Figure 4 materials-12-03433-f004:**
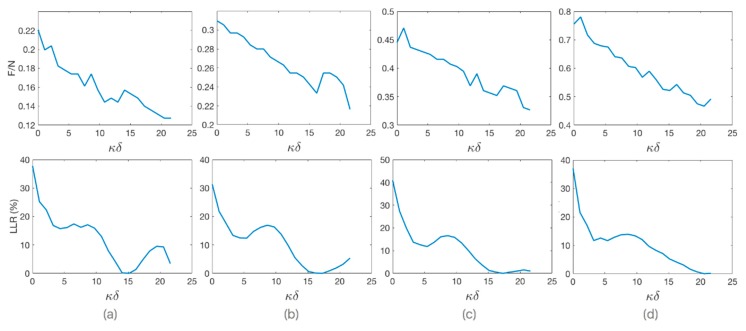
(Top) Value of F/N and (Bottom) relative loss of lateral resolution, both, vs. stand-off given in number of wavelengths (*κ**δ*) for 445 kHz PDMS-based axicon lens with different angles. (**a**) *φ* = 140°; (**b**) *φ* = 150°; (**c**) *φ* = 160°; (**d**) *φ* = 170°.

**Figure 5 materials-12-03433-f005:**
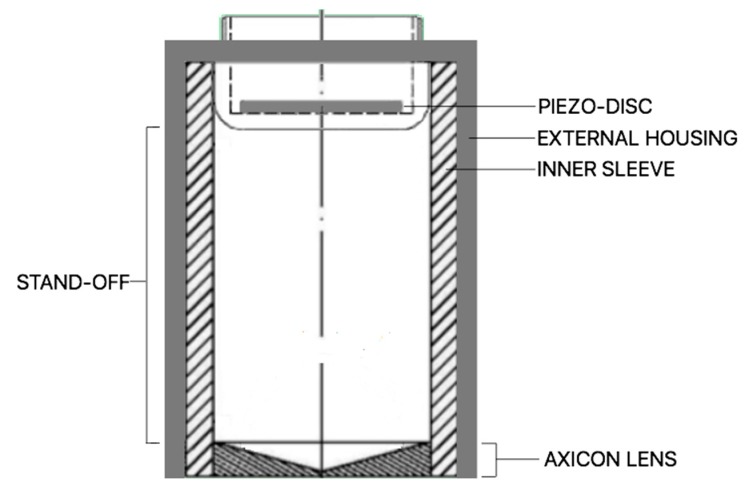
Housing of the axicon lens.

**Figure 6 materials-12-03433-f006:**
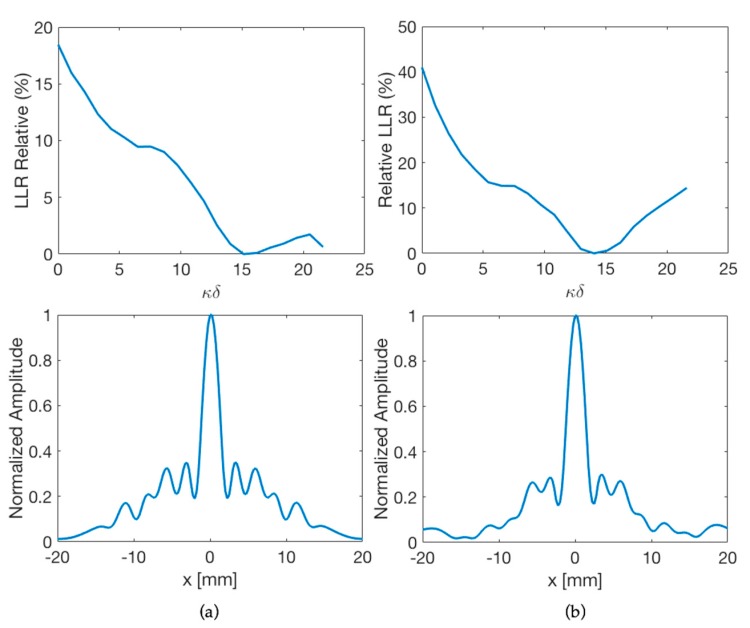
Outer case effect for 445 kHz PDMS-based axicon lens with cone angle *φ* = 130°. (Top) Relative loss of lateral resolution vs. stand-off given in number of wavelengths (*κ**δ*) and (Bottom) normalized pressure amplitudes of the lateral beam profile. (**a**) Lens housing with reflectivity of 0.8 dB down; (**b**) lens housing with reflectivity of 40 dB down.

**Figure 7 materials-12-03433-f007:**
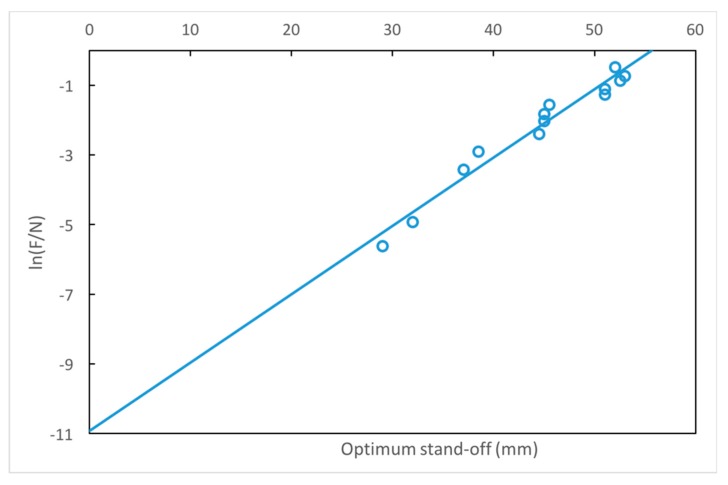
Illustration of the relation between the axicon lens optimum stand-off *δ* and the ratio of F/N. The value of *δ* estimated by linear regression is indicated to obtain the highest lateral resolution for different lens angles.

**Figure 8 materials-12-03433-f008:**
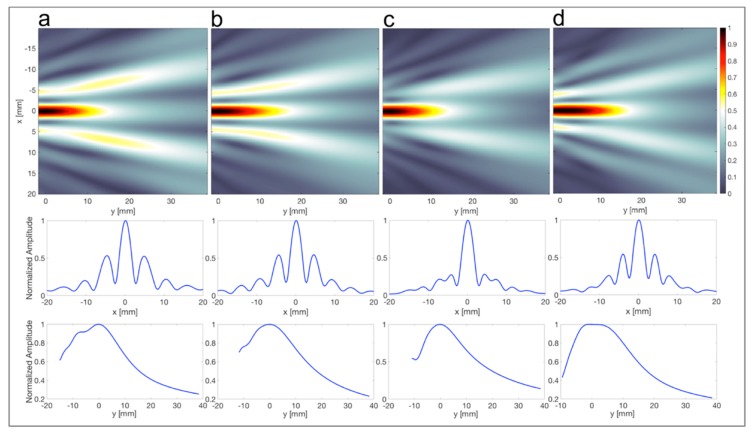
445 kHz-Cigar-shaped acoustic focus for different settings of the stand-off. (**a**) *δ* = 0 mm; (**b**) *δ* = 20.5 mm; (**c**) *δ* = 34 mm; (**d**) *δ* = 45.25 mm. (Top) Focusing behavior of the PDMS-based 144° axicon lens with four different values of *δ*. (Middle) Normalized pressure amplitudes of the lateral beam profile. (Bottom) Normalized pressure amplitudes of the axial beam profile, 0 mm indicates the focus.

**Figure 9 materials-12-03433-f009:**
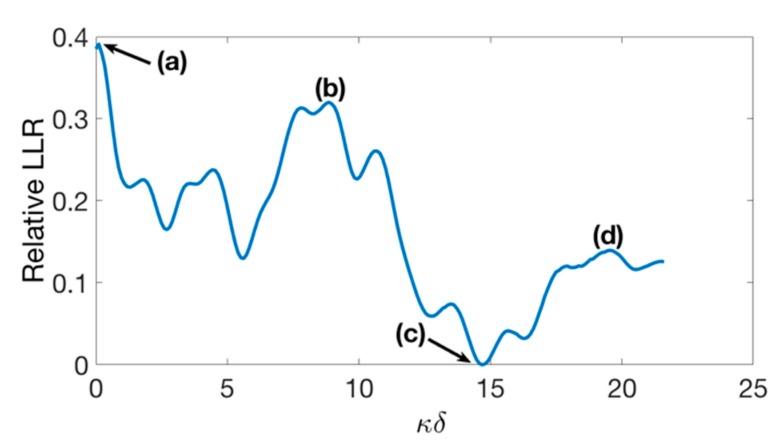
Stand-off, given in number of wavelengths, vs. relative loss of lateral resolution for 445 kHz PDMS-based 144° axicon lens. (**a**) *δ* = 0 mm; (**b**) *δ* = 20.5 mm; (**c**) *δ* = 34 mm; and (**d**) *δ* = 45.25 mm.

**Figure 10 materials-12-03433-f010:**
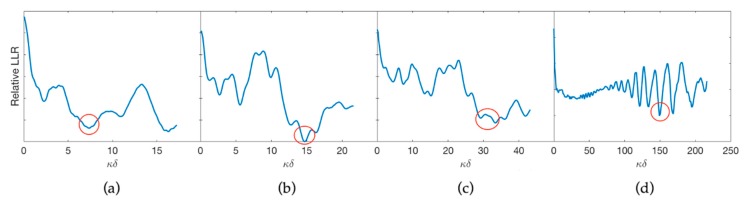
Stand-off given in number of wavelengths vs. relative loss of lateral resolution for different frequencies. The value of *κ**δ* for the optimum stand-off (red circle) is directly proportional to the frequency. (**a**) f = 0.2225 MHz; (**b**) f = 0.445 MHz; (**c**) f = 0.890 MHz; (**d**) f = 4.45 MHz.

**Figure 11 materials-12-03433-f011:**
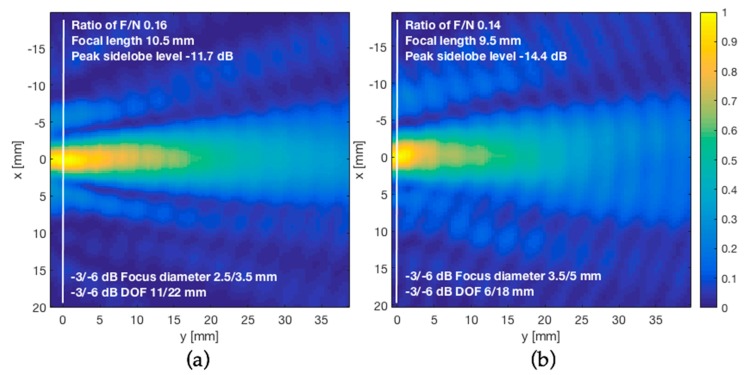
Ultrasound can be focused through human skull phantom. Experimental measurements of acoustic pressure field emitted from a 445 kHz axicon lens. (**a**) Free space without skull; (**b**) after transcranial transmission through skull bone phantom. White line indicates the focus.

**Table 1 materials-12-03433-t001:** Compressional and shear speed, attenuation, and density of Clear Med610 material.

Comp. Speed (ms^−1^)	Shear Speed (ms^−1^)	Absorption (dB cm^−1^)	Density (kg m^−3^)
2495 ± 8	1081 ± 31	3.70 ± 0.1	1180

**Table 2 materials-12-03433-t002:** Specifications of the axicon lens characterized.

Parameter	Value
Transducer frequency (f)	0.445 MHz
Transducer diameter (D)	28 mm
Transducer near field in water (N)	64.75 mm (−6 dB)
Ratio of F/N (F/N)	0.16
Axicon lens angle (*φ*)	144°
Focal length (F)	10.5 mm
Depth of focus (DOF)	11 mm (−3 dB)/22 mm (−6 dB)
Focus diameter (d_F_)	2.5 mm (−3 dB)/3.5 mm (−6 dB)
Insertion loss (IL)	8.4 dB
Stand-off (*δ*)	30 mm
Encapsulation Echo Reduction (ER)	−12 dB

**Table 3 materials-12-03433-t003:** Influence of the thickness skull in the focus of axicon lenses.

Thickness Skull (ts)	0.75 mm ts < λ/4	1.25 mm ts < λ/4	1.5 mm λ/4 < ts < λ/2	2.5 mm λ/4 < ts < λ/2	3 mm λ/2 < ts < λ	5 mm λ/2 < ts < λ	6 mm ts > λ
F (mm)	10.75	11	11	9.75	8.25	9.5	10.5
d_F_ (mm)	3.5	3.5	3.5	5	4	4	4
DOF (mm)	17	18	18	17.5	17	17.5	17
SLL (dB)	−8.2	−10.3	−11.5	−8.4	−8.7	−9.9	−9.5

**Table 4 materials-12-03433-t004:** Comparison of simulated and scanned acoustic beam properties obtained using a configuration of 445 kHz transducer with a 144° Epoxy/PDMS lens, through the phantom of 5 mm thickness, separated 2 mm from the lens.

Beam Properties	Simulated	Scanned
d_F_ (mm)	4	5
DOF (mm)	17.5	18
SLL (dB)	−9.9	−14.4

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
