# Peer review of "Ultrasound Axicon: Systematic Approach to Optimize Focusing Resolution through Human Skull Bone"

_materials, 2019, doi:10.3390/ma12203433_

Round 1
Reviewer 1 Report
Would the authors describe more about the parietal bone phantom used in the experimental validation? Is it a home made or commercial available phantom? What are the geometry and acoustic properties? Overall, I am curious how much it will change the sound beams, if I were to simply use a typical spherical focusing transducer?
Reviewer 2 Report
The paper presented a systematic way to optimize the resolution of axicon lenses. However, I failed to understand the relevance of the topic in the journal materials. No material characterization was presented. I think this paper is much suitable in some Bio/optical journal rather than materials
Author Response
We appreciate very much the comments of the reviewer.
Reviewer 3 Report
This paper describes the numerical and experimental results for the focusing properties of axicon ultrasound lens. In the simulations, the optimum separation between the ultrasound transducer and the axicon lens (stand-off) was investigated, and in the experiments, the effect of a human skull on the ultrasound field focused by the axicon lens was investigated.
I think that this paper needs major revision.
1) I think that the validity of the simulation method is doubtful. Should shear stresses should be considered in the ultrasound simulations in solids?
2) The study purpose should be clarified, the results suitable for the purpose should be shown, and the results should be discussed.
- Page 1, Abstract: The results should be clearly described.
- Page 3, lines 104-123: The contents of simulations and experiments were different. If the experiments are performed in order to demonstrate the validity of the simulations, they should be the same so that their results can be directly compared. In particular, is it not appropriate that the frequency of 0.445 MHz (Table 2) in the experiments was different from 0.515 MHz (Table A1) in the simulations?
- Page 5, Sec. 3.2:
- Page 8, lines 209-211: I don't know which simulated results the experimental results were in good coincidence with.
- Page 9, 4. Discussion: It should be clear which results are being discussed.
3) I think that the explanations are totally not sufficient.
- Page 4, Figure 2: It should be specified which is the probe, phantom, and hydrophone. What is the difference between “probe” and “hydrophone” in the caption? Are the lens and transducer not shown in the photo?
- Pages 4-5, Table 2 and Figure 3: How to calculate the values of F/N? Theoretical calculation?
- Page 5, lines 136-142: The basis for this explanation should be shown. Was it derived from simulated results or referred from the literature?
- Page 5, lines 144-145: Please explain the definition of LLR.
- Page 5, line 146: Please explain the definition of the average sound speed (how it is averaged).
- Page 5, Sec. 3.2: Please explain the encapsulation.
- Page 5, Eq. 8: Please show the determination coefficient and p-value.
- Page 12, Figure A1: Please specify what the color in the figure represents.
4) Other comments
- Page 5, line 147: Should "ratio of" be removed from “ratio of F/N"?
- Page 9, line 236: Should either "will affect" or "will depend on" be removed?
- Page 10-12, Appendix A: I think that It is redundant, and Table A1 and Figure 1 had better be written in Chap. 2.
Reviewer 4 Report
General comment: This work proposed a method to create focused ultrasound field by focusing the sound field generated by plane transducer using axicon lens. The numerical simulations help to find the optimized parameters for the proposed system design. The sound field testing using hydrophone showed the performance of the proposed method by comparing the cases with and without a human skull phantom.
Specific comments:
The second sentence of Abstract is not correct in grammar, which cause confusion. This reviewer suggests the authors to label the ultrasound probe, phantom, and the hydrophone in the photo to facilitate the reading process of the readers. Concerning the brain skull phantom. Is the material in the phantom is uniform or non-uniform? The bone of human skull is a porous like structure, which can cause great diffraction effects. If the phantom material uniform, how much off the obtained measurements in Fig. 10 (b) is compared to the real skull bone? If the phantom is non-uniform, how similar it is to the real skull bone? In Fig. 7, what is the relationship between to lateral axes: y axis, and depth axis? Are they referring to the same physical axis? If not, what is the difference? Why there is negative depth? What is the 0 mm in depth referred to? Concerning the difference of sound field properties. What is the physical cause for 40% increase of focus diameter and 18% decrease of depth of focus? What are the consequences for the targeted applications: brain stimulation and ultrasound therapy? What are the state-of-the-art of the requirements sound field property for focused ultrasound application in brain stimulation and ultrasound therapy? How well do the sound field properties produced by the proposed method satisfy those requirements? What are the difference in performance between the conventional spherically focused transducer and the transducer generated by the proposed method? The advantage of the proposed method, claimed by the authors, is to create a shorter focal length and suppress the near field. How much improvement does this proposed method have compared with existing technology such as spherically focused transducers? In this work, the authors said that “The skull is not an obstacle for focusing ultrasound with axion lenses”. For the real application, in which the skin is capable observe a lot of nearfield energy through heating generation, how well the proposed method can perform compared with spherically focused transducers?
Round 2
Reviewer 2 Report
The paper is good for publication.
Author Response
Reviewer's comments are appreciated.
Reviewer 3 Report
It is difficult for me to sufficiently understand the contents of this study, although I believe that the interesting results are contained. I think that the reason is because the structure of this manuscript is not good, and the purpose, methods, results, discussion, and conclusions are not logically organized. I recommend to revise the structure.
Reviewer 4 Report
Some of the punctuations of the Abstract are problematic, and should be corrected. The authors did not respond to this reviewer's comments on the state-of-the-art of the sound field in brain stimulation. It is ok if this work is not claimed to used for brain stimulation, but it is NOT ok if this work aims at brain stimulation. The reason is simple: If the current work does not have any advantage over the state-of-the-art, it is useless. But if the current is better than the current state-of-the-art in some way, please clarify it.
